# AlignFlow: Learning from multiple domains via normalizing flows

**Aditya Grover,**[*] **Christopher Chute**[*]**, Rui Shu, Zhangjie Cao, Stefano Ermon**
Computer Science Department
Stanford University

## Abstract

The goal of unpaired cross-domain translation is to learn useful mappings between two domains, given only unpaired sets of datapoints from these domains. While this formulation is highly underconstrained, recent work has shown that it is possible to learn mappings useful for downstream tasks by encouraging approximate *cycle consistency* in the mappings between the two domains (Zhu et al., 2017a). In this work, we propose AlignFlow, a framework for unpaired cross-domain translation that ensures *exact* cycle consistency in the learned mappings. Our framework uses a normalizing flow model to specify a *single* invertible mapping between the two domains. In contrast to prior works in cycle-consistent translations, we can learn AlignFlow via adversarial training, maximum likelihood estimation, or a hybrid of the two methods. Theoretically, we derive consistency results for AlignFlow which guarantee recovery of desirable mappings under suitable assumptions. Empirically, AlignFlow demonstrates significant improvements over relevant baselines on image-to-image translation and unsupervised domain adaptation tasks on benchmark datasets.

## 1 Introduction

Given data from two domains, cross-domain translation refers to the task of learning a mapping from one domain to another, such as translating text across two languages or image colorization. This ability to learn a meaningful alignment between two domains has a broad range of applications across machine learning, including relational learning (Kim et al., 2017), domain adaptation (Taigman et al., 2016; Hoffman et al., 2017; Bousmalis et al., 2017), image and video translation for computer vision (Isola et al., 2017; Wang et al., 2018), and machine translation for natural language processing (Lample et al., 2017).

Broadly, there are two learning paradigms for cross-domain translation: paired and unpaired. In *paired* cross-domain translation, we assume access to pairs of datapoints across the two domains, e.g., black and white images and their respective colorizations. However, paired data can be expensive to obtain or may not even exist, as in neural style transfer (Gatys et al., 2015) where the goal is to translate across the works of two artists that typically do not exhibit a direct correspondence.

*Unpaired* cross-domain translation tackles this regime where paired data is not available and learns an alignment between two domains given only unpaired sets of datapoints from the domains. Formally, we seek to learn a joint distribution over two domains, say A and B, given samples only from the marginal distributions over A and B. CycleGAN (Zhu et al., 2017a), a highly successful approach to this problem, learns a pair of conditional generative models, say $G_{A \to B}$ and $G_{B \to A}$, to match the marginal distributions over A and B via an adversarial objective (Goodfellow et al., 2014). The marginal matching constraints alone are insufficient to learn the desired joint distribution, both in theory and practice. To further constrain the problem, an additional desideratum is imposed in the form of *cycle-consistency*. That is, given any datapoint A $= a$, the cycle-consistency term in the learning objective prefers mappings $G_{A \to B}$ and $G_{B \to A}$ such that $G_{B \to A}(G_{A \to B}(a)) \approx a$. Symmetrically, cycle-consistency in the reverse direction implies $G_{A \to B}(G_{B \to A}(b)) \approx b$ for all datapoints B $= b$. Intuitively, this encourages the learning of *approximately bijective* mappings.

---

[*]Equal contribution.

While empirically effective, the CycleGAN objective only imposes a soft cycle-consistency penalty and provides no guarantee that $G_{A\to B}$ and $G_{B\to A}$ are true inverses of each other. A natural question, then, is whether the cycle-consistency objective can be replaced with a *single*, invertible model $G_{A\to B}$. Drawing inspiration from the literature on invertible generative models (Rezende and Mohamed, 2015; Dinh et al., 2014; 2017; Kingma and Dhariwal, 2018), we propose AlignFlow, a learning framework for cross-domain translations which uses normalizing flow models to represent the mappings. In AlignFlow, we compose a pair of invertible flow models $G_{Z\to A}$ and $G_{Z\to B}$, to represent the mapping $G_{A\to B} = G_{Z\to B} \circ G_{Z\to A}^{-1}$. Here, Z is a shared latent space between the two domains. Since composition of invertible mappings preserves invertibility, the mapping $G_{A\to B}$ is invertible and the reverse mapping from B → A is simply given as $G_{B\to A} = G_{A\to B}^{-1}$. Hence, AlignFlow guarantees *exact* cycle-consistency by design and simplifies the standard CycleGAN learning objective by learning a single, invertible mapping.

Furthermore, AlignFlow provides flexibility in specifying the training objective. In addition to adversarial training, we can also specify a prior distribution over the latent variables Z and train the two component models $G_{Z\to B}$ and $G_{Z\to A}$ via maximum likelihood estimation (MLE). MLE is statistically efficient, exhibits stable training dynamics, and can have a regularizing effect when used in conjunction with adversarial training of invertible generative models (Grover et al., 2018).

## 2 PRELIMINARIES

In this section, we discuss the necessary background and notation on generative adversarial networks, normalizing flows, and cross-domain translations using CycleGANs. Unless explicitly stated otherwise, we assume probability distributions admit absolutely continuous densities on a suitable reference measure. We use uppercase notation X, Y, Z to denote random variables, and lowercase notation $x, y, z$ to denote specific values in the italicized corresponding sample spaces $\mathcal{X}, \mathcal{Y}, \mathcal{Z}$.

### 2.1 GENERATIVE ADVERSARIAL NETWORKS

A generative adversarial network (GAN) is a latent variable model which specifies a deterministic mapping $h : \mathcal{Z} \to \mathcal{X}$ between a set of latent variables Z and a set of observed variables X (Goodfellow et al., 2014). In order to sample from GANs, we need a prior density over Z that permits efficient sampling. A GAN generator can also be conditional, where the conditioning is on another set of observed variables (and optionally the latent variables Z as before) (Mirza and Osindero, 2014).

A GAN is trained via adversarial training, wherein the generator $h$ plays a minimax game with an auxiliary critic C. The goal of the critic $C : \mathcal{X} \to \mathbb{R}$ is to distinguish real samples from the observed dataset with samples generated via $h$. The generator, on the other hand, tries to generate samples that can maximally confuse the critic. Many learning objectives have been proposed for adversarial training, including those based on f-divergences (Nowozin et al., 2016), Wasserstein Distance (Arjovsky et al., 2017), and maximum mean discrepancy (Li et al., 2017). The generator and the critic are both parameterized by deep neural networks and learned via alternating gradient-based optimization. Because adversarial training only requires samples from the generative model, it can be used to train generative models with intractable or ill-defined likelihoods (Mohamed and Lakshminarayanan, 2016). In practice, such likelihood-free methods give excellent performance on sampling-based tasks unlike the alternative maximum likelihood estimation-based training criteria for learning generative models. However, these models are harder to train due to the alternating minimax optimization and suffer from issues such as mode collapse (Goodfellow, 2016).

### 2.2 NORMALIZING FLOWS

Normalizing flows represent a latent variable generative model that specifies an *invertible* mapping $h : \mathcal{Z} \to \mathcal{X}$ between a set of latent variables Z and a set of observed variables X. Let $p_X$ and $p_Z$ denote the marginal densities defined by the model over $\mathcal{X}$ and $\mathcal{Z}$ respectively. Using the change-of-variables formula, the marginal densities can be related as:

$$p_X(x) = p_Z(z) \left| \det \frac{\partial h^{-1}}{\partial X} \right|_{X=x} \tag{1}$$

where $z = h^{-1}(x)$ due to the invertibility constraints. Here, the second term on the RHS corresponds to the absolute value of the determinant of the Jacobian of the inverse transformation and signifies the shrinkage/expansion in volume when translating across the two sample spaces.

For evaluating likelihoods via the change-of-variables formula, we require efficient and tractable evaluation of the prior density, the inverse transformation $h^{-1}$, and the determinant of its Jacobian of $h^{-1}$. To draw a sample from this model, we perform ancestral sampling, i.e., we first sample a latent vector $z \sim p_Z(z)$ and obtain the sampled vector as given by $x = h(z)$. This requires the ability to efficiently: (1) sample from the prior density and (2) evaluate the forward transformation $h$. Many transformations parameterized by deep neural networks that satisfy one or more of these criteria have been proposed in the recent literature on normalizing flows, e.g., NICE (Dinh et al., 2014) and Autoregressive Flows (Kingma et al., 2016; Papamakarios et al., 2017). By suitable design of transformations, both likelihood evaluation and sampling can be performed efficiently, as in Real-NVP (Dinh et al., 2017). Consequently, a flow model can be trained efficiently via maximum likelihood estimation as well as likelihood-free adversarial training (Grover et al., 2018).

## 2.3 DOMAIN TRANSLATIONS VIA CYCLEGAN

Consider two multi-variate random variables A and B with domains specified as $\mathcal{A} \subseteq \mathbb{R}^n$ and $\mathcal{B} \subseteq \mathbb{R}^n$ respectively. Let $p^*_{A,B}$ denote the joint distribution over these two variables. In the unpaired cross-domain translation setting, we are given access to a finite datasets $\mathcal{D}_A$ and and $\mathcal{D}_B$, sampled independently from the two unknown corresponding (marginal) data distributions $p^*_A$ and $p^*_B$ respectively. Using these datasets, the goal is to learn the conditional distributions $p^*_{A|B}$ and $p^*_{B|A}$. Without any paired data, the problem is underconstrained (even in the limit of infinite paired data) since the conditionals can only be derived from $p^*_{A,B}$, but we only have data sampled from the marginal densities. To address this issue, CycleGAN introduced additional constraints that have proven to be empirically effective in learning mappings that are useful for downstream tasks. We now proceed by describing the CycleGAN framework.

If we assume the conditional distributions for A|B and B|A are deterministic, the conditionals can alternatively be represented as cross-domain mappings $G_{A \to B} : \mathcal{A} \to \mathcal{B}$ and $G_{B \to A} : \mathcal{B} \to \mathcal{A}$. A CycleGAN uses a pair of conditional GANs to translate data from two domains (Zhu et al., 2017a). It consists of the following components:

1. A conditional GAN $G_{A \to B} : \mathcal{A} \to \mathcal{B}$ that takes as input data from domain $\mathcal{A}$ and maps it to domain $\mathcal{B}$. The mapping $G_{A \to B}$ is learned adversarially with the help of a critic $C_B : \mathcal{B} \to \mathbb{R}$ trained to distinguish between real and synthetic data (generated via $G_{A \to B}$) from domain $\mathcal{B}$.

2. Symmetrically, a conditional GAN $G_{B \to A} : \mathcal{B} \to \mathcal{A}$ and a critic $C_A : \mathcal{A} \to \mathbb{R}$ for adversarial learning of the reverse mapping from $\mathcal{B}$ to $\mathcal{A}$.

Any suitable GAN loss can be substituted in the above objective, e.g., Wasserstein GAN (Arjovsky et al., 2017). For the standard cross-entropy based GAN loss, the critic outputs a probability of a datapoint being real and optimizes the following objective:

$$\mathcal{L}_{GAN}(C_A, G_{B \to A})$$
$$:= E_{a \sim p^*_A}[\log C_A(a)] + E_{b \sim p^*_B}[\log(1 - C_A(G_{B \to A}(b)))]. \tag{2}$$

Additionally, semantically meaningful mappings can be learned via a pair of conditional GANs $G_{A \to B}$ and $G_{B \to A}$ that are encouraged to be *cycle consistent*. Cycle consistency encourages the data translated from domain $\mathcal{A}$ to $\mathcal{B}$ via $G_{A \to B}$ to be mapped back to the original datapoints in $\mathcal{A}$ via $G_{B \to A}$. That is, $G_{B \to A}(G_{A \to B}(a)) \approx a$ for all $a \in \mathcal{A}$. Formally, the cycle-consistency loss for translation from A to B and back is defined as:

$$\mathcal{L}_{Cycle}(G_{B \to A}, G_{A \to B})$$
$$:= E_{a \sim p^*_A}[\|G_{B \to A}(G_{A \to B}(a)) - a\|_1] \tag{3}$$

Symmetrically, an additional cycle consistency term $\mathcal{L}_{Cycle}(G_{A \to B}, G_{B \to A})$ in the reverse direction encourages $G_{A \to B}(G_{B \to A}(b)) \approx b$ for all $b \in \mathcal{B}$.

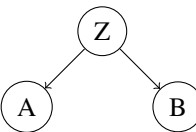

Figure 1: Bayesian network for AlignFlow.

The full objective optimized by a CycleGAN is given as:

$$\mathcal{L}_{\text{CycleGAN}}(G_{\text{B}\rightarrow\text{A}}, C_{\text{A}}, G_{\text{A}\rightarrow\text{B}}, C_{\text{B}}; \lambda_{\text{A}\rightarrow\text{B}}, \lambda_{\text{B}\rightarrow\text{A}})$$
$$:= \mathcal{L}_{\text{GAN}}(C_{\text{A}}, G_{\text{B}\rightarrow\text{A}}) + \mathcal{L}_{\text{GAN}}(C_{\text{B}}, G_{\text{A}\rightarrow\text{B}})$$
$$+ \lambda_{\text{A}\rightarrow\text{B}}\mathcal{L}_{\text{Cycle}}(G_{\text{B}\rightarrow\text{A}}, G_{\text{A}\rightarrow\text{B}})$$
$$+ \lambda_{\text{B}\rightarrow\text{A}}\mathcal{L}_{\text{Cycle}}(G_{\text{A}\rightarrow\text{B}}, G_{\text{B}\rightarrow\text{A}}) \tag{4}$$

where $\lambda_{\text{A}\rightarrow\text{B}}$ and $\lambda_{\text{B}\rightarrow\text{A}}$ are hyperparameters controlling the relative strength of the cycle consistent terms. The objective is minimized w.r.t. $G_{\text{B}\rightarrow\text{A}}, G_{\text{A}\rightarrow\text{B}}$ and maximized w.r.t. $C_{\text{A}}, C_{\text{B}}$. In practice, the expectations w.r.t. $p_{\text{A}}^*$ and $p_{\text{B}}^*$ in the individual loss terms are approximated via the datasets $\mathcal{D}_{\text{A}}$ and $\mathcal{D}_{\text{B}}$ respectively.

The use of cycle consistency has indeed been shown empirically to be a good inductive bias for learning cross-domain translations. However, it necessitates a careful design of the loss function that could involve a trade-off between the adversarial training and cycle consistency terms in the objective in Eq. 4. To stabilize training and achieve good empirical performance, Zhu et al. (2017a) proposes a range of techniques such as the use of an *identity loss* in the above objective.

## 3 THE ALIGNFLOW FRAMEWORK

In this section, we present the AlignFlow framework for learning cross-domain translations between two domains $\mathcal{A}$ and $\mathcal{B}$. We will first discuss the model representation, followed by the learning and inference procedures for AlignFlow. Finally, we will present a theoretical result analyzing the proposed framework.

### 3.1 REPRESENTATION

We will use a graphical model to represent the relationships between the domains to be translated. Consider a Bayesian network between two sets of observed random variables A and B with domains $\mathcal{A}$ and $\mathcal{B}$ respectively along with a parent set of unobserved random variable Z with domain $\mathcal{Z}$. The network is illustrated in Figure 1.

The latent variables Z indicate a shared feature space between the observed variables A and B, which will be exploited later for efficient learning and inference. While Z is unobserved, we assume a prior density $p_{\text{Z}}$ over these variables, such as an isotropic Gaussian. The marginal densities over A and B are not known, and will be learned using the unpaired data from the two domains.

Finally, to specify the joint distribution between these sets of variables, we constrain the relationship between A and Z, and B and Z to be invertible. That is, we specify mappings $G_{\text{Z}\rightarrow\text{A}}$ and $G_{\text{Z}\rightarrow\text{B}}$ such that the respective inverses $G_{\text{A}\rightarrow\text{Z}} = G_{\text{Z}\rightarrow\text{A}}^{-1}$ and $G_{\text{B}\rightarrow\text{Z}} = G_{\text{Z}\rightarrow\text{B}}^{-1}$ exist. In the proposed AlignFlow framework, we specify the cross-domain mappings as the composition of two invertible mappings:

$$G_{\text{A}\rightarrow\text{B}} = G_{\text{Z}\rightarrow\text{B}} \circ G_{\text{A}\rightarrow\text{Z}} \tag{5}$$
$$G_{\text{B}\rightarrow\text{A}} = G_{\text{Z}\rightarrow\text{A}} \circ G_{\text{B}\rightarrow\text{Z}}. \tag{6}$$

Since composition of invertible mappings is invertible, both $G_{\text{A}\rightarrow\text{B}}$ and $G_{\text{B}\rightarrow\text{A}}$ are invertible. In fact, it is straightforward to observe that $G_{\text{A}\rightarrow\text{B}}$ and $G_{\text{B}\rightarrow\text{A}}$ are inverses of each other:

$$G_{\text{A}\rightarrow\text{B}}^{-1} = (G_{\text{Z}\rightarrow\text{B}} \circ G_{\text{A}\rightarrow\text{Z}})^{-1} = G_{\text{A}\rightarrow\text{Z}}^{-1} \circ G_{\text{Z}\rightarrow\text{B}}^{-1}$$
$$= G_{\text{Z}\rightarrow\text{A}} \circ G_{\text{B}\rightarrow\text{Z}} = G_{\text{B}\rightarrow\text{A}}. \tag{7}$$

Hence, AlignFlow only needs to specify the forward mapping from one domain to another. The corresponding mapping in the reverse direction is simply given by the inverse of the forward mapping. Such a choice permits increased flexibility in specifying learning objectives and performing efficient inference, which we discuss next.

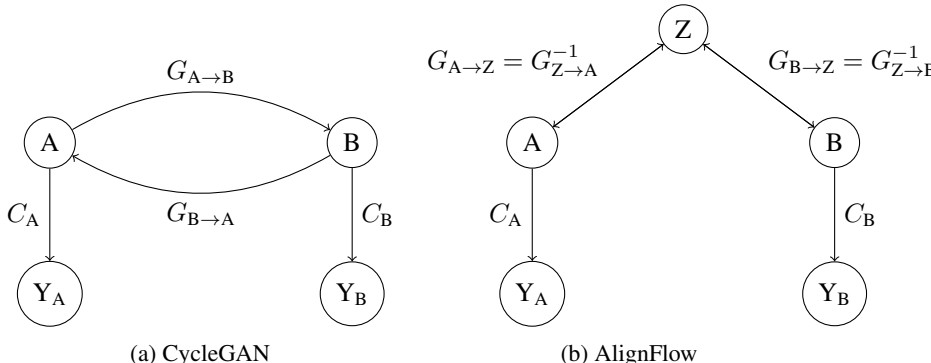

(a) CycleGAN                               (b) AlignFlow

Figure 2: CycleGAN v.s. AlignFlow. Unlike CycleGAN, AlignFlow specifies a single, invertible mapping $G_{A \to Z} \circ G_{B \to Z}^{-1}$ that is exactly cycle-consistent, represents a shared latent space Z between the two domains, and can be trained via both adversarial training and exact maximum likelihood estimation. Double-headed arrows in AlignFlow denote invertible mapping. $Y_A$ and $Y_B$ are random variables denoting the output of the critics used for adversarial training.

## 3.2 LEARNING ALGORITHMS & OBJECTIVES

From a probabilistic standpoint, the cross-domain translation problem requires us to learn a conditional distribution $p_{A|B}^*$ over A and B given data sampled from the corresponding marginals $p_A^*$ and $p_B^*$.

We now discuss two methods to learn a mapping from $\mathcal{B} \to \mathcal{A}$ such that the resulting marginal distribution over A, denoted as $p_A$ is close to $p_A^*$. Unless mentioned otherwise, all our results that hold for a particular domain $\mathcal{A}$ will have a natural counterpart for the domain $\mathcal{B}$, by the symmetrical nature of the problem setup and the AlignFlow framework.

**Adversarial Training.** A flow model representation permits efficient ancestral sampling. Hence, a likelihood-free framework to learn the conditional mapping from $\mathcal{B}$ to $\mathcal{A}$ is to perform adversarial training similar to a GAN. That is, we introduce a critic $C_A$ that plays a minimax game with the generator mapping $G_{B \to A}$. The critic $C_A$ distinguishes real samples $a \sim p_A^*$ with the generated samples $G_{B \to A}(b)$ for $b \sim p_B^*$. An example GAN loss is illustrated in Eq. 2.

Alternatively if our goal is to only learn a generative model with the marginal density close to $p_A^*$, then we can choose to simply learn the mapping $G_{Z \to A}$. As shown in Grover et al. (2018), the mapping $G_{Z \to A}$ along with an easy-to-sample prior density $p_Z$ itself specifies a latent variable model that can learned via an adversarial training objective, similar to the one illustrated in Eq. 2 or any other GAN loss.

**Maximum Likelihood Estimation.** Flow models can also be trained via maximum likelihood estimation (MLE). Hence, an MLE objective for learning the mapping $G_{Z \to A}$ maximizes the likelihood of the dataset $\mathcal{D}_A$:

$$\mathcal{L}_{\text{MLE}}(G_{Z \to A}) := E_{a \sim p_A^*}[\log p_A(a)] \tag{8}$$

$$\text{where } p_A(a) = p_Z(G_{A \to Z}^{-1}(a)) \left| \det \frac{\partial G_{A \to Z}^{-1}}{\partial A} \right|_{A=a}.$$

As in the previous cases, the expectation w.r.t. $p_A^*$ is approximated via Monte Carlo averaging over the dataset $\mathcal{D}_A$. Besides efficient evaluation of the inverse transformations and its Jacobian, this objective additionally requires a prior with a tractable density, e.g. an isotropic Gaussian.

**Cycle-consistency.** So far, we have only discussed objectives for modeling the marginal density over A (and symmetrical learning objectives exist for B). However, as discussed previously, the marginal densities alone do not guarantee learning a mapping that is useful for downstream tasks.

Cycle consistency, as proposed in CycleGAN (Zhu et al., 2017a), is a highly effective learning objective that encourages learning of meaningful cross-domain mappings. For AlignFlow, we observe that cycle consistency is *exactly* satisfied. Formally, we have the following result:

**Proposition 1.** *Let $\mathcal{G}$ denote the class of invertible mappings represented by an arbitrary AlignFlow architecture. For any $G_{B \to A} \in \mathcal{G}$, we have:*

$$\mathcal{L}_{\text{Cycle}}(G_{B \to A}, G_{A \to B}) = 0 \tag{9}$$
$$\mathcal{L}_{\text{Cycle}}(G_{A \to B}, G_{B \to A}) = 0 \tag{10}$$

*where $G_{A \to B} = G_{B \to A}^{-1}$ by design.*

The proposition follows directly from the invertible design of the AlignFlow framework (Eq. 7).

**Overall objective.** In AlignFlow, we optimize a combination of the adversarial learning objective and the maximum likelihood objective.

$$\begin{aligned}
&\mathcal{L}_{\text{AlignFlow}}(G_{B \to A}, C_A, C_B; \lambda_A, \lambda_B) \\
&:= \mathcal{L}_{\text{GAN}}(C_A, G_{B \to A}) + \mathcal{L}_{\text{GAN}}(C_B, G_{A \to B}) \\
&\quad - \lambda_A \mathcal{L}_{\text{MLE}}(G_{Z \to A}) - \lambda_B \mathcal{L}_{\text{MLE}}(G_{Z \to B})
\end{aligned} \tag{11}$$

where $\lambda_A \geq 0$ and $\lambda_B \geq 0$ are hyperparameters that reflect the strength of the MLE terms for domains A and B respectively. The AlignFlow objective is minimized w.r.t. the parameters of the generator $G_{A \to B}$ and maximized w.r.t. parameters of the critics $C_A$ and $C_B$. Notice that we have expressed $\mathcal{L}_{\text{AlignFlow}}$ as a function of the critics $C_A, C_B$ and only $G_{B \to A}$ since the latter also encompasses the other parametric functions appearing in the objective ($G_{A \to B}, G_{Z \to A}, G_{Z \to B}$) via the invertibility constraints in Eqs. 5-7. For different choices of $\lambda_A$ and $\lambda_B$, we cover the following three cases:

1. **Adversarial training only:** For $\lambda_A = \lambda_B = 0$, we recover the CycleGAN objective in Eq. 4, with the additional benefits of exact cycle consistency and a single invertible generator. In this case, the prior over Z plays no role in learning.

2. **MLE only:** On the other extreme for large values of $\lambda_A, \lambda_B$ such that $\lambda_A = \lambda_B \to \infty$, we can perform pure maximum likelihood training to learn the invertible generator. Here, the critics $C_A, C_B$ play no role since the adversarial training terms are ignored in Eq. 11.

3. **Hybrid:** For any finite, non-zero value of $\lambda_A, \lambda_B$, we obtain a hybrid objective where both the adversarial and MLE terms are accounted for during learning.

## 3.3 INFERENCE

AlignFlow can be used for both conditional and unconditional sampling at test time. For conditional sampling, we are given a datapoint $b \in \mathcal{B}$ and we can draw the corresponding cross-domain translation in domain $\mathcal{A}$ via the mapping $G_{B \to A}$.

For unconditional sampling, we require $\lambda_A \neq 0$ since doing so will activate the use of the prior $p_Z$ via the MLE terms in the learning objective. Thereafter, we can obtain samples by first drawing $z \sim p_Z$ and then applying the mapping $G_{Z \to A}$ to $z$. Furthermore, the same $z$ can be mapped to domain $\mathcal{B}$ via $G_{Z \to B}$. Hence, we can sample paired data $(G_{Z \to A}(z), G_{Z \to B}(z))$ given $z \sim p_Z$.

## 3.4 COMPARISON WITH CYCLEGAN

AlignFlow differs from CycleGAN with respect to the model family as well as the learning algorithm and inference capabilities. We illustrate and compare both models in Figure 2. CycleGAN parameterizes two independent mappings $G_{A \to B}$ and $G_{B \to A}$, whereas AlignFlow only specifies a single, invertible mapping. Learning in a CycleGAN is restricted to an adversarial training objective along with a cycle-consistent loss term, whereas AlignFlow is exactly consistent and can be trained via adversarial learning, MLE, or a hybrid. Finally, inference in CycleGAN is restricted to conditional sampling since it does not involve any latent variables Z with easy-to-sample prior densities. As described previously, AlignFlow permits both conditional and unconditional sampling.

# 4 THEORETICAL ANALYSIS

For finite non-zero values of $\lambda_A$ and $\lambda_B$, the AlignFlow objective consists of three parametric models: one generator $G_{B \to A} \in \mathcal{G}$, and two critics $C_A \in \mathcal{C}_A, C_B \in \mathcal{C}_B$. Here, $\mathcal{G}, \mathcal{C}_A, \mathcal{C}_B$ denote model families specified e.g., via deep neural network based architectures. In this section, we analyze the optimal solutions to these parameterized models within well-specified model families.

## 4.1 MARGINAL-CONSISTENCY

Our first result characterizes the conditions under which the optimal generators exhibit *marginal-consistency* for the data distributions defined over the domains $\mathcal{A}$ and $\mathcal{B}$.

**Definition 1.** *Let $p_{X,Y}$ denote the joint distribution between two domains $\mathcal{X}$ and $\mathcal{Y}$. An invertible mapping $G_{Y \to X} : \mathcal{Y} \to \mathcal{X}$ is marginally-consistent w.r.t. two arbitrary distributions $(p_X, p_Y)$ iff for all $x \in \mathcal{X}, y \in \mathcal{Y}$:*

$$p_X(x) = \begin{cases} p_Y(y) \left| \det \frac{\partial G_{Y \to X}^{-1}}{\partial Y} \right|_{Y=y}, & \text{if } x = G_{Y \to X}(y) \\ 0, & \text{otherwise.} \end{cases} \tag{12}$$

Next, we show that AlignFlow is marginally-consistent for well-specified model families.

**Lemma 1.** *Let $\mathcal{G}_A$ and $\mathcal{G}_B$ denote the class of invertible mappings represented by the AlignFlow architecture for mapping $Z \to A$ and $Z \to B$. For a given choice of prior distribution $p_Z$, if there exist mappings $G_{Z \to A}^* \in \mathcal{G}_A, G_{Z \to B}^* \in \mathcal{G}_B$ that are marginally consistent w.r.t. $(p_A^*, p_Z)$ and $(p_B^*, p_Z)$ respectively, then the mapping $G_{B \to A}^* = G_{Z \to A}^* \circ G_{Z \to B}^{*^{-1}}$ is marginally-consistent w.r.t. $(p_A^*, p_B^*)$.*

The result follows directly from Definition 1 and change-of-variables applied to the mapping $G_{B \to A}^* = G_{Z \to A}^* \circ G_{Z \to B}^{*^{-1}}$.

**Theorem 1.** *Assume that the model families for the critics $C_A : \mathcal{A} \to [0,1]$ and $C_B : \mathcal{B} \to [0,1]$ are the set of all measurable functions for the cross-entropy GAN objective. Then, $G_{B \to A}^*$ (as defined in Lemma 1) globally minimizes the AlignFlow objective in Eq. 11 for any value of $\lambda_A \geq 0, \lambda_B \geq 0$.*

*Proof.* See Appendix A.1. □

Note that marginally-consistent mappings w.r.t. a target data distribution and a prior density need not be unique. While an invertible model family mitigates the underconstrained nature of the problem, it does not provably eliminate it. We provide some non-identifiable constructions in Appendix A.3 and leave the exploration of additional constraints that guarantee identifiability to future work.

## 4.2 OPTIMAL CRITICS

Unlike standard adversarial training of an unconditional normalizing flow model (Grover et al., 2018; Danihelka et al., 2017), the AlignFlow model involves two critics. Here, we are interested in characterizing the dependence of the optimal critics for a given invertible mapping $G_{A \to B}$. Consider the AlignFlow framework where the GAN loss terms in Eq. 11 are specified via the cross-entropy objective in Eq. 2. For this model, we can relate the optimal critics using the following result.

**Theorem 2.** *Let $p_A^*$ and $p_B^*$ denote the true data densities for domains $\mathcal{A}$ and $\mathcal{B}$ respectively. Let $C_A^*$ and $C_B^*$ denote the optimal critics for the AlignFlow objective with the cross-entropy GAN loss for any fixed choice of the invertible mapping $G_{A \to B}$. Then, we have for any $a \in \mathcal{A}$:*

$$C_A^*(a) = \frac{C_B^*(b) p_A^*(a)}{p_A^*(a) + p_B^*(b)(1 - C_B^*(b)) \left| \det \frac{\partial G_{A \to B}^{-1}}{\partial A} \right|_{A=a}} \tag{13}$$

*where $b = G_{A \to B}(a)$.*

*Proof.* See Appendix A.2. □

In essence, the above result shows that the optimal critic for one domain, w.l.o.g. say A, can be directly obtained via the optimal critic of another domain B for any choice of the invertible mapping $G_{A \to B}$, assuming one were given access to the data marginals $p_A^*$ and $p_B^*$.

Table 1: Mean Squared Error (MSE) comparing CycleGAN and varaints of AlignFlow on paired test sets. MSE is computed pixelwise after normalizing images to $(-1, 1)$.

| Dataset | Model | MSE (A $\rightarrow$ B) | MSE (B $\rightarrow$ A) |
|---|---|---|---|
| Facades | CycleGAN | 0.7129 | 0.3286 |
| | AlignFlow (Adversarial only) | 0.6727 | 0.2679 |
| | AlignFlow (Hybrid) | **0.5801** | **0.2512** |
| | AlignFlow (MLE only) | 0.9014 | 0.5960 |
| Maps | CycleGAN | 0.0245 | 0.0953 |
| | AlignFlow (Adversarial only) | 0.0385 | 0.1123 |
| | AlignFlow (Hybrid) | **0.0209** | **0.0897** |
| | AlignFlow (MLE only) | 0.0452 | 0.1746 |
| CityScapes | CycleGAN | 0.1252 | **0.1200** |
| | AlignFlow (Adversarial only) | 0.2569 | 0.2196 |
| | AlignFlow (Hybrid) | **0.1130** | 0.1462 |
| | AlignFlow (MLE only) | 0.2526 | 0.2272 |

Table 2: Test classification accuracies for domain adaptation from source→target. The source only and target only models directly use classifiers trained on the source and target datasets respectively.

| Model | MNIST→USPS | USPS→MNIST | SVHN→MNIST |
|---|---|---|---|
| source only | $82.2 \pm 0.8$ | $69.6 \pm 3.8$ | $67.1 \pm 0.6$ |
| ADDA (Tzeng et al., 2017) | $89.4 \pm 0.2$ | $90.1 \pm 0.8$ | $76.0 \pm 1.8$ |
| CyCADA + CycleGAN | $95.6 \pm 0.2$ | $96.5 \pm 0.1$ | $90.4 \pm 0.4$ |
| CyCADA + AlignFlow | $\mathbf{96.2 \pm 0.2}$ | $\mathbf{96.7 \pm 0.1}$ | $\mathbf{91.0 \pm 0.3}$ |
| target only | $96.3 \pm 0.1$ | $99.2 \pm 0.1$ | $99.2 \pm 0.1$ |

## 5 EXPERIMENTS

In this section, we empirically evaluate AlignFlow for image-to-image translation and unsupervised domain adaptation. For both these tasks, the most relevant baseline is CycleGAN. Extensions to CycleGAN that are complementary to our work are excluded for comparison to ensure a controlled evaluation. We discuss these extensions in detail in Section 6. In all our experiments, we specify the AlignFlow architecture based on the invertible transformations introduced in Real-NVP (Dinh et al., 2017). For experimental details beyond those stated below, we refer the reader to Appendix B.

### 5.1 IMAGE-TO-IMAGE TRANSLATION

We evaluate AlignFlow on three image-to-image translation datasets used by Zhu et al. (2017a): Facades, Maps, and CityScapes (Cordts et al., 2016). These datasets are chosen because they provide aligned image pairs, so one can quantitatively evaluate unpaired image-to-image translation models via a distance metric such as mean squared error (MSE) between generated examples and the corresponding ground truth. Note that we restrict ourselves to unpaired translation, so the pairing information is omitted during training and only used for evaluation.

While MSE can have limitations, we follow prior evaluation protocols and report the MSE for translations on the test sets after cross-validation of hyperparameters in Table 1. For hybrid models, we set $\lambda_A = \lambda_B$. We observe that while learning AlignFlow via adversarial training or MLE alone is not as competitive as CycleGAN, hybrid training of AlignFlow significantly outperforms CycleGAN in almost all cases. Specifically, we observe that MLE alone typically performs worse than adversarial training, but together both these objectives seem to have a regularizing effect on each other. Qualitative evaluation of the reconstructions for all datasets is deferred to Appendix B.

## 5.2 UNSUPERVISED DOMAIN ADAPTATION

The setup for unsupervised domain adaptation (Saenko et al., 2010) is as follows. We are given data from two related domains: a source and a target domain. For the source, we have access to both the input datapoints and their labels. For the target, we are only provided with input datapoints without any labels. Using the available data, the goal is to learn a classifier for the target domain.

A variety of algorithms have been proposed for the above task which seek to match pixel-level or feature-level distributions across the two domains. One such model relevant to this experiment is Cycle-Consistent Domain Adaptation (CyCADA) (Hoffman et al., 2017). CyCADA first learns a cross-domain translation mapping from source to target domain via CycleGAN. This mapping is used to stylize the source dataset into the target domain, which is then subject to additional feature-level and semantic consistency losses for learning the target domain classifier (Ganin and Lempitsky, 2014; Tzeng et al., 2017). A full description of CyCADA is beyond the scope of discussion of this work; we direct the reader to Hoffman et al. (2017) for further details.

In this experiment, we seek to assess the usefulness of AlignFlow for domain adaptation in the CyCADA framework. We evaluate the same pairs of source and target datasets as in Hoffman et al. (2017): MNIST (LeCun et al., 1998), USPS (Hull, 1994), SVHN (Netzer et al., 2011), which are all image datasets of handwritten digits with 10 classes. Instead of training a source-to-target and a target-to-source generator with a cycle-consistency loss term, we train AlignFlow with only the GAN-based loss in the target direction. In Table 2, we see that CyCADA based models perform better in two out of three adaptation settings when used in conjunction with AlignFlow.

## 6 RELATED WORK

A key assumption in unsupervised domain alignment is the existence of a deterministic or stochastic mapping $G_{A \to B}$ such that the distribution of B matches that of $G_{A \to B}(A)$, and vice versa. This assumption can be incorporated as a marginal distribution-matching constraint into the objective using an adversarially-trained GAN critic (Goodfellow et al., 2014). However, this objective is under-constrained. To partially mitigate this issue, CycleGAN (Zhu et al., 2017a), DiscoGAN (Kim et al., 2017), and DualGAN (Yi et al., 2017) added an approximate cycle-consistency constraint, by encouraging $G_{B \to A} \circ G_{A \to B}$ and $G_{A \to B} \circ G_{B \to A}$ to behave like identity functions on domains A and B respectively. While cycle-consistency is empirically very effective, alternatives based on variational autoencoders that do not require either cycles or adversarial training have also been proposed recently (Hoshen, 2018; Hoshen and Wolf, 2018).

In a parallel line of work, CoGAN (Liu and Tuzel, 2016) and UNIT (Liu et al., 2017) demonstrated the efficacy of adding a shared-space constraint, where two decoders (decoding into domains A and B respectively) share the same latent space. These works have since been extended to enable one-to-many mappings (Huang et al., 2018a; Zhu et al., 2017b) as well as multi-domain alignment (Choi et al., 2018). Our work focuses on the one-to-one unsupervised domain alignment setting. In contrast to previous models, AlignFlow leverages both a shared latent space and *exact* cycle-consistency. To our knowledge, AlignFlow provides the first demonstration that invertible models can be used successfully in lieu of the cycle-consistency objective. Furthermore, AlignFlow allows the incorporation of exact maximum likelihood training, which we demonstrated to induce a meaningful shared latent space that is amenable to interpolation.

To enforce exact cycle-consistency, we leverage the growing literature on invertible generative models. Dinh et al. (2014) proposed a class of volume-preserving invertible neural networks (NICE) that uses the change of variables formulation to enable exact maximum likelihood training. Real-NVP (Dinh et al., 2017) and Flow++ (Ho et al., 2019) extend this line of work by allowing volume transformations and additional architectural considerations. Glow (Kingma and Dhariwal, 2018) further builds upon this by incorporating invertible $1 \times 1$ convolutions. We note that additional lines of work based on autoregressive flows (Kingma et al., 2016; Papamakarios et al., 2017; Huang et al., 2018b), ordinary differential equations-based flows (Chen et al., 2018; Grathwohl et al., 2018), and planar flows (Berg et al., 2018) have shown improvements in specific scenarios. For fast inversion, our work makes use of the Real-NVP model, and we leave extensions of this model in the unsupervised domain alignment setting as future work.

## 7 CONCLUSION & FUTURE WORK

In this work, we presented AlignFlow, a learning framework for cross-domain translations based on normalizing flow models. The use of normalizing flow models is an attractive choice for several reasons we highlight: it guarantees exact cycle-consistency via a single cross-domain mapping, learns a shared latent space across two domains, and permits a flexible training objective which is a hybrid of terms corresponding to adversarial training and exact maximum likelihood estimation. Theoretically, we derived conditions under which the AlignFlow model learns marginals that are consistent with the underlying data distributions. Finally, our empirical evaluation demonstrated significant gains on the tasks of image-to-image translation and unsupervised domain adaptation, along with an increase in inference capabilities due to the use of invertible models, e.g., paired interpolations in the latent space for two domains.

In the future, we would like to consider extensions of AlignFlow to learning stochastic, multimodal mappings (Zhu et al., 2017b) and translations across more than two domains (Choi et al., 2018). In spite of strong empirical results in domain alignments in the last few years, a well-established theory explaining such results is lacking. With a handle on model likelihoods and exact invertibility for inference, we are optimistic that AlignFlow can potentially aid the development of such a theory and characterize structure that leads to provably identifiable recovery of cross-domain mappings. Exploring the latent space of AlignFlow from a manifold learning perspective to domain alignment (Cui et al., 2014) is also an interesting direction for future research.

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

# APPENDICES

# A    PROOFS OF THEORETICAL RESULTS

## A.1    PROOF OF THEOREM 1

*Proof.* Since the maximum likelihood estimate minimizes the KL divergence between the data and model distributions, the optimal value for $\mathcal{L}_{\text{MLE}}(G_{Z \to A})$ is attained at a marginally-consistent mapping, say $G^*_{Z \to A}$. Symmetrically, there exists a marginally-consistent mapping $G^*_{Z \to B}$ that optimizes $\mathcal{L}_{\text{MLE}}(G_{Z \to B})$.

From Theorem 1 of Goodfellow et al. (2014), we know that the cross-entropy GAN objective $\mathcal{L}_{\text{GAN}}(C_A, G_{B \to A})$ is globally minimized when $p_A = p^*_A$ and critic is Bayes optimal. Further, from Lemma 1, we know that $G^*_{B \to A}$ is marginally-consistent w.r.t. $(p^*_A, p^*_B)$. Hence, $G^*_{B \to A}$ globally minimizes $\mathcal{L}_{\text{GAN}}(C_A, G_{B \to A})$. Symmetrically, $G^*_{A \to B} = G^{*^{-1}}_{B \to A}$ globally minimizes $\mathcal{L}_{\text{GAN}}(C_B, G_{A \to B})$.

Since $G^*_{B \to A} = G^*_{Z \to A} \circ G^{*^{-1}}_{Z \to B}$ globally optimizes all the individual loss terms in the AlignFlow objective in Eq. 11, it globally optimizes the overall objective for any value of $\lambda_A \geq 0, \lambda_B \geq 0$.

$\square$

## A.2    PROOF OF THEOREM 2

*Proof.* First, we note that only the GAN loss terms depend on $C_A$ and $C_B$. Hence, the MLE terms are constants for a fixed $G_{B \to A}$ and hence, can be ignored for deriving the optimal critics. Next, for any GAN trained with the cross-entropy loss as specified in Eq 2, we know that the Bayes optimal critic $C^*_A$ prediction for any $a \in \mathcal{A}$ is given as:

$$C^*_A(a) = \frac{p^*_A(a)}{p^*_A(a) + p_A(a)} \tag{14}$$

See Proposition 1 in Goodfellow et al. (2014) for a proof.

We can relate the densities $p_A(a)$ and $p_B(b)$ via the change of variables as:

$$p_A(a) = p_B(b) \left| \det \frac{\partial G^{-1}_{A \to B}}{\partial A} \right|_{A=a} \tag{15}$$

where $b = G_{A \to B}(a)$.

Substituting the expression for density of $p_A(a)$ from Eq. 15 in Eq. 14, we get:

$$C^*_A(a) = \frac{p^*_A(a)}{p^*_A(a) + p_B(b) \left| \det \frac{\partial G^{-1}_{A \to B}}{\partial A} \right|_{A=a}} \tag{16}$$

where $b = G_{A \to B}(a)$.

Symmetrically, using Proposition 1 in Goodfellow et al. (2014) we have the Bayes optimal critic $C^*_B$ for any $b \in \mathcal{B}$ given as:

$$C^*_B(b) = \frac{p^*_B(b)}{p^*_B(b) + p_B(b)}. \tag{17}$$

Rearranging terms in Eq. 17, we have:

$$p_B(b) = p^*_B(b) \left( \frac{1}{C^*_B(b)} - 1 \right) \tag{18}$$

for any $b \in \mathcal{B}$.

Substituting the expression for density of $p_B(b)$ from Eq. 18 in Eq. 16, we get:

$$C^*_A(a) = \frac{C^*_B(b) p^*_A(a)}{p^*_A(a) + p^*_B(b)(1 - C^*_B(b)) \left| \det \frac{\partial G^{-1}_{A \to B}}{\partial A} \right|_{A=a}} \tag{19}$$

where $b = G_{A \to B}(a)$.

$\square$

### A.3  NON-IDENTIFIABILITY

As discussed, marginal consistency along with invertibility can only reduce the underconstrained nature of the unpaired cross-domain translation problem, but not completely eliminate it. In the following result, we identify one such class of non-identifiable model families for the MLE-only objective of AlignFlow ($\lambda_A = \infty, \lambda_B = \infty$). We will need the following definitions.

**Definition 2.** *Let $\mathcal{S}_n$ denotes the symmetric group on $n$ dimensional permutation matrices. A function class for the cross-domain mappings $\mathcal{G}$ is closed under permutations iff for all $G_{B \to A} \in \mathcal{G}, S \in \mathcal{S}_n$, we have $G_{B \to A} \circ S \in \mathcal{G}$.*

**Definition 3.** *A density $p_X$ is symmetric iff for all $x \in \mathcal{X} \subseteq \mathbb{R}^n, S \in \mathcal{S}_n$, we have $p_X(x) = p_X(Sx)$.*

Examples of distributions with symmetric densities include the isotropic Gaussian and Laplacian distributions.

**Proposition 2.** *Consider the case where $G^*_{B \to A} \in \mathcal{G}$, and $\mathcal{G}$ is closed under permutations. For a symmetric prior $p_Z$ (e.g., isotropic Gaussian), there exists an optimal solution $G^\dagger_{B \to A} \in \mathcal{G}$ to the AlignFlow objective (Eq. 11) for $\lambda_A = \lambda_B = \infty$ such that $G^\dagger_{B \to A} \neq G^*_{B \to A}$.*

*Proof.* We will prove the proposition via contradiction. That is, let's assume that $G^*_{B \to A}$ is a unique solution for the AlignFlow objective for $\lambda_A = \lambda_B = \infty$ (Eq. 11). Now, consider an alternate mapping $G^\dagger_{B \to A} = G^*_{B \to A} S$ for an arbitrary non-identity permutation matrix $S \neq I$ in the symmetric group.

As before, we note that $G^*_{B \to A} = G^*_{Z \to A} \circ G^{*^{-1}}_{Z \to B}$ and $G^\dagger_{B \to A} = G^\dagger_{Z \to A} \circ G^{\dagger^{-1}}_{Z \to B}$ due to the invertibility constraints in Eqs. 5-7. Since permutation matrices are invertible and so is $G^*_{B \to A}$, their composition given by $G^\dagger_{B \to A}$ is also invertible. Further, since $\mathcal{G}$ is closed under permutation and $G^*_{B \to A} \in \mathcal{G}$, we also have $G^\dagger_{B \to A} \in \mathcal{G}$.

Next, we note that the inverse of a permutation matrix is also a permutation matrix. Since the prior is assumed to be symmetric and a a transformation specified by a permutation matrix is volume-preserving (i.e., $\det(S) = 1$ for all $S \in \mathcal{S}_n$), we can use the change-of-variables formula in Eq. 1 to get:

$$\mathcal{L}_{\text{MLE}}(G^*_{Z \to A}) = \mathcal{L}_{\text{MLE}}(G^\dagger_{Z \to A}) \tag{20}$$

$$\mathcal{L}_{\text{MLE}}(G^*_{Z \to B}) = \mathcal{L}_{\text{MLE}}(G^\dagger_{Z \to B}). \tag{21}$$

Noting that $G^*_{B \to A} = G^*_{Z \to A} \circ G^{*^{-1}}_{Z \to B}$ and $G^\dagger_{B \to A} = G^\dagger_{Z \to A} \circ G^{\dagger^{-1}}_{Z \to B}$ due to the invertibility constraints in Eqs. 5-7, we can substitute the above equations in Eq. 11. When $\lambda_A = \lambda_B = \infty$, for any choice of $C_A, C_B$ we have:

$$\mathcal{L}_{\text{AlignFlow}}(G^*_{B \to A}, C_A, C_B, \lambda_A = \infty, \lambda_B = \infty)$$
$$= \mathcal{L}_{\text{AlignFlow}}(G^\dagger_{B \to A}, C_A, C_B, \lambda_A = \infty, \lambda_B = \infty). \tag{22}$$

The above equation implies that $G^\dagger_{B \to A}$ is also an optimal solution to the AlignFlow objective in Eq. 11 for $\lambda_A = \lambda_B = \infty$. Thus, we arrive at a contradiction since $G^*_{B \to A}$ is not the unique maximizer. Hence, proved. $\square$

The above construction suggests that MLE-only training can fail to identify the optimal mapping corresponding to the joint distribution $p^*_{A,B}$ even if it lies within the mappings represented via the family represented via the AlignFlow architecture. Failure modes due to non-identifiability could also potentially arise for adversarial and hybrid training. Empirically, we find that while MLE-only training gives poor performance for cross-domain translations, the hybrid and adversarial training objectives are much more effective, which suggests that these objectives are less susceptible to identifiability issues in recovering the true mapping.

# B EXPERIMENT DETAILS

## B.1 IMAGE-TO-IMAGE TRANSLATION

We use the standard training, validation, and test splits for each dataset. For datasets which do not provide a validation set (*e.g.,* Facades and CityScapes), we randomly hold out a portion of the training set with the same number of images as the test set. We train each model for 200 epochs with a fixed learning rate of $2 \cdot 10^{-4}$ for the first 100 epochs, followed by a linear decay schedule for 100 epochs from the initial learning rate to 0. We use the Adam (Kingma and Ba, 2014) optimizer with $\beta_1 = 0.5$ and $\beta_2 = 0.999$, and for AlignFlow we apply weight normalization (Salimans and Kingma, 2016) of $5 \cdot 10^{-5}$ to the generator's parameters. When training with an MLE objective, we apply gradient clipping with a maximum gradient norm of 10. Scaling flow models to higher dimensionality is an active area of research; for this work we resized the images to $64 \times 64$ for Cityscapes and Maps, and $128 \times 128$ for Facades. We use a batch size of 4 images per GPU and trained over 4 GPUs in parallel. For CycleGAN results, all hyperparameters are adopted from Zhu et al. (2017a).

For MLE/Hybrid models, we used an isotropic Gaussian prior. We use the following flow architecture to parameterize $G_{Z \to A}$ and $G_{Z \to B}$:

> Scale[Input: 32x32x3, Output: 16x16x6x2]
> $\to$ 3x CheckerboardCoupling[Channels: 32, Blocks: 4]
> $\to$ 3x ChannelwiseCoupling[Channels: 64, Blocks: 4]
> $\to$ Squeeze&Split[Input: 32x32x3, Output: 16x16x6x2]
> Scale[Input: 16x16x6, Output: 8x8x12x2]
> $\to$ 3x CheckerboardCoupling[Channels: 64, Blocks: 4]
> $\to$ 3x ChannelwiseCoupling[Channels: 128, Blocks: 4]
> $\to$ Squeeze&Split[Input: 16x16x6, Output: 8x8x12x2]
> Scale[Input: 8x8x12, Output: 4x4x24x2]
> $\to$ 3x CheckerboardCoupling[Channels: 128, Blocks: 4]
> $\to$ 3x ChannelwiseCoupling[Channels: 256, Blocks: 4]
> $\to$ Squeeze&Split[Input: 8x8x12, Output: 4x4x24x2]
> Scale[Input: 4x4x24, Output: 4x4x24]
> $\to$ 4x CheckerboardCoupling[Channels: 256, Blocks: 4]

where CheckerboardCoupling and ChannelwiseCoupling are affine coupling layers with checkerboard and channelwise masking, respectively, and where Squeeze&Split first trades spatial extent for channels by turning each $4 \times 4 \times 1$ subvolume into a $1 \times 1 \times 4$ subvolume, and then splits the volume along the last dimension and sends half of the features directly to the latent space. See Dinh et al. (2017) for more details. Within each affine coupling layer, we parametrize the scale and translate factors using a ResNet (He et al., 2016) architecture with the specified number of channels and residual blocks. We additionally use activation normalization Kingma and Dhariwal (2018) before each coupling layer.

## B.2 UNSUPERVISED DOMAIN ADAPTATION

We use the same training, validation and test splits of MNIST, USPS, and SVHN digit datasets as in CyCADA (Hoffman et al., 2017). For all datasets, images are resized to $32 \times 32$ as in CyCADA. We employ the pixel-level and feature-level adaptation training pipeline as in CyCADA but replace the CycleGAN-based image translation network with the AlignFlow. The architectures for imposing semantic consistency and feature adaptation are the same as the ones used for CyCADA. The architecture and hyperparameter tuning protocol was consistent with the one used for image-to-image translations using AlignFlow. For the hyperparameters of feature-level domain adaptation post the image translations, we adopted the optimal hyperparameter settings from ADDA (Tzeng et al., 2017).

