# OpenReview forum: "AlignFlow: Cycle Consistent Learning from Multiple Domains via Normalizing Flows"
_ICLR.cc/2019/Workshop/DeepGenStruct — DeepGenStruct 2019_

### Official Review · AnonReviewer2 · 2019-04-06
**Taking cycle consistency to a new level**

**Rating:** 4
**Confidence:** 3

**Review:**

This paper shows how to use flow models for unpaired image to image translation, by leveraging the invertbility of flows, by sharing a common latent space between two models which map from this latent space into the two domains of interest.

Pros:
  * some nice guarantees due to the invertbility properties of flows
  * good empirical results - showing some of the baselines in the cycleGAN paper.
  * building on top of prior work which uses adversarial training to train flows.

Cons:
   * RNVPs require more computation to achieve high quality samples, due to the local structure of the model and the reliance on checkerboard and channel alternating patterns. There is no discussion on the model size in AlignFlow and that required by CycleGAN.
  * lacking some of the most impressive results from cyclegan.

Question for the authors:
  * in my experience, RNVPs are quite fiddly to train. I expect that when adversarial training is added to the mix, things get even more fiddly. What did you do to stabilize your model?
  * how sensitive is the model to the choice of 'lambda_a and \lambda_b? I could not find any discussion on that - nor the values for \lambda_a and \lambda_b in the paper.

---

### Official Review · AnonReviewer1 · 2019-04-18
**Great paper**

**Rating:** 4
**Confidence:** 3

**Review:**

The paper proposes AlignFlow, an efficient way of implementing cycle consistency principle using invertible flows. The paper is clearly written and I really enjoyed reading it!
Pros:
- Clever combination of existing ideas (use invertible mappings rather than encoder-decoder pairs in cycleGAN)
- simple to implement
- works well in practice

This paper proposes a related idea and might be worth discussing:
Invertible Autoencoder for domain adaptation https://arxiv.org/pdf/1802.06869.pdf

---

### Decision · Program_Chairs · 2019-04-19
**Acceptance Decision**

Accept